# Abiotic $N_2$ reduction in submarine hydrothermal systems could quickly fertilize prebiotic oceans

Liheng Sun[1,2], Kan Li [2], Zhen Sun[1,3,4] ✉, Yunying Zhang[1] & Long Li [2] ✉

$NH_3$ or $NH_4^+$ is an essential component of the abiotic synthesis of organic compounds for the origin of life and an efficient greenhouse gas to address the faint young Sun paradox on the early Earth. Sustainable $NH_3$ or $NH_4^+$ on the $N_2$-dominated prebiotic Earth's surface requires potent abiotic $N_2$ reduction (ANR) in hydrothermal systems, which has not been detected in the geological record despite numerous laboratory demonstrations. Here we report high concentrations and extreme $^{15}N$ depletions of $NH_4^+$ in hydrothermal veins in oceanic crusts drilled from the South China Sea basin. Our data indicate that abundant $^{15}N$-depleted $NH_4^+$ was produced by ANR in deep fluid but progressively overprinted by $^{15}N$-enriched biogenic $NH_4^+$ toward the surface. Modeling suggests that ANR could supply up to $9.0 – 10.8 \times 10^{10}$ mol·year$^{-1}$ $NH_4^+$ to global oceans, which is minor to the large nitrogen inventory in modern oceans, but could quickly fertilize the oceans and supply $NH_3$ to the atmosphere in the prebiotic Earth.

Given the important role of abiotic $N_2$ reduction (ANR) in early Earth's geological nitrogen (N) cycle, particularly in providing the essential compound of $NH_3$ (or its dissolved equivalent $NH_4^+$) for the origin of life in submarine hydrothermal vents[1–3] and for establishing a warm and habitable environment under dimmer solar luminosity (i.e., the faint young Sun paradox)[4,5] on the $N_2$-dominant early Earth surface[6], this process has been intensively tested by laboratory experiments. The experimental results demonstrated that ANR could be catalyzed by a variety of naturally occurring minerals and rocks (e.g., FeS, Fe-Ni alloy, green rust, magnetite, peridotite) under submarine hydrothermal conditions[7–18]. However, ANR has not been convincingly detected in field samples, particularly modern submarine hydrothermal vent fluids that have been collected for study. One of the major responsible factors is the overprinting by $NH_4^+$ generated in shallow hydrothermal systems from biologically processed N sources such as organic matter, dissolved $NO_3^-$ and $NH_4^+$ in seawater, and dissolved organic N and $NH_4^+$ in pore water of sediment. For example, when a sediment cover exists near hydrothermal systems, elevated temperature (T) condition can induce decomposition of organic matter and/or $NH_4^+$ desorption from clays[19]. In addition, dissolved $NO_3^-$ in seawater can be effectively reduced (even abiotically) in shallow hydrothermal systems as low as 24 °C[17,18]. These processes can contribute remarkable amounts of $NH_4^+$ to increase the $NH_4^+$ concentrations of shallow hydrothermal fluids to more than an order of magnitude higher than the ambient seawater $NH_4^+$ concentration (~1 μM)[20]. Because these $NH_4^+$ components are all derived from surface N sources in shallow localities, they are referred to as surface $NH_4^+$ hereafter. Due to the similar N isotopic signatures of these surface N sources, e.g., +3‰ to +8‰ for dissolved $NO_3^-$ in seawater[21], +2‰ to +10‰ for marine organic matter/sediments[22] and similar range (with high value up to +17‰) for dissolved organic N and $NH_4^+$ in interstitial water[23], it is difficult to distinguish between $NH_4^+$ derived from these surface sources. Regardless, such $^{15}N$-enriched surface $NH_4^+$ can effectively overprint the $^{15}N$-depleted signal from deep source (e.g., −5‰ for the upper mantle)[24,25] in shallow hydrothermal fluids.

[1]State Key Laboratory of Tropical Oceanography, South China Sea Institute of Oceanology, Chinese Academy of Sciences, Guangzhou, China. [2]Department of Earth and Atmospheric Sciences, University of Alberta, Edmonton, AB, Canada. [3]Key Laboratory of Marine Mineral Resources, Ministry of Natural Resources, Guangzhou Marine Geological Survey, China Geological Survey, Guangzhou, China. [4]China-Pakistan Joint Research Center on Earth Sciences, CAS-HEC, Islamabad, Pakistan. ✉e-mail: sun_zhen2024@126.com; long4@ualberta.ca

A better geological proxy for detecting an ANR signal would be the overlooked hydrothermal veins deposited from focused flow of deep hydrothermal fluids, for two reasons. Firstly, although deep hydrothermal fluids could be derived from deeply circulated seawater, surface $NH_4^+$ could be progressively consumed (by $NH_4^+$ assimilation into alteration minerals in oceanic crust[26–31]) along seawater circulation pathway into depths. Consequently, the deep hydrothermal fluids should undergo minimal impact from surface $NH_4^+$ and have the best chance to expose $NH_4^+$ produced by ANR. Secondly, studies on field samples[32,33], laboratory experiments[34] and theoretical calculations[35] suggest that $NH_4^+$ can substitute $K^+$ and $Na^+$ in silicate minerals. Thus $NH_4^+$ in deep fluids can partially partition into $K^+$- and $Na^+$-bearing vein minerals (e.g., plagioclase, epidote, chlorite) upon their deposition. Once fixed in the structure of vein minerals, $NH_4^+$ can be well protected from subsequent low-T disturbance. As a result, vein minerals deposited in the focused flow channel of deep fluids can best reveal the deep-fluid $NH_4^+$ signature and have the best chance to disclose the ANR signal (if there is any).

Despite recent advance in characterizing the $NH_4^+$ signature of hydrothermally altered oceanic crust (i.e., seafloor basalts, sheeted dikes and gabbros, and serpentinized peridotites)[26–31], $NH_4^+$ in hydrothermal veins in oceanic crust has been rarely examined by far. International Ocean Discovery Program (IODP) Expeditions 367 and 368 drilled into the 16–32 million-year-old oceanic crusts in the South China Sea basin[36] (Fig. 1; Supplementary Information). Recovered mid-ocean ridge basalts (MORB) from Hole U1502B show an E-MORB affinity (Supplementary Fig. 2). All these rocks have been altered at various degrees and contain abundant hydrothermal veins with thickness from submillimeter to a few millimeters (Supplementary Fig. 3). The mineral assemblage of hydrothermal veins is dominated by albite and quartz with variable amounts of Fe-Mg-Ca carbonates, chlorite, epidote, pyrite, and Fe−Mn hydroxides ("Methods"; Supplementary Data 1 and Supplementary Fig. 4), which were precipitated from relatively high-T fluids (200−300 °C; Supplementary Information)[36–38]. The hydrothermal fluids, as determined from trace elements and radiogenic isotopes of vein minerals, were a mixture of modified seawater (after reaction with MORB) and deep magmatic fluid[38]. Here we report the N concentrations and isotope compositions of these vein samples, in comparison with those of their hosting altered MORB, to constrain

the N cycle pathways and estimate the $NH_4^+$ flux in the deep oceanic hydrothermal systems.

## Results and discussion

### N enrichment in altered MORB and veins

The altered MORB from Hole U1502B have bulk-rock N concentrations from 18 to 40 µg/g (Fig. 1), which are much higher than that of fresh MORB (<2 µg/g) but still fall in the upper end of the N concentration range of global altered MORB (2–48 µg/g)[29], indicating that the alteration-induced N enrichment in these basalts is similar to those in global altered MORB. The $\delta^{15}N$ values of the U1502B basalts (−7.6‰ to +0.2‰; Supplementary Data 2) are lower than the values of the surface source[21–23] and thus suggest that the secondary N came from not only seawater/sedimentary source but also a $^{15}N$-depleted source. However, similar to the low-$\delta^{15}N$ hydrothermally altered basalts from ODP Sites 801 and 1149[27] and DSDP Site 417[29], no good correlation was observed between the bulk-rock concentrations of N and any other elements (Supplementary Data 2; Supplementary Fig. 5). This can be attributed to the variable hydrothermal conditions over their alteration history, e.g., $NH_4^+$ content, temperature, and secondary mineral assemblage (see detailed discussion in Yu et al.[19]). For example, high-T (>~300 °C) alteration minerals (e.g., amphibole and albite) have lower $NH_4^+$-hosting capabilities than low-T (<~300 °C) alteration minerals (e.g., clays)[19,30]. Some minerals (e.g., clays) even have multiple $NH_4^+$-hosting sites with high $NH_4^+$-hosting capacity in interlayer sites and low $NH_4^+$-hosting capacity in surface and edge sites[19]. Furthermore, oceanic basalts might experience low-T microbial alteration by lithochemotrophic organisms[39] which could also contribute some biological N in rocks, but has not been quantified so far. All these complexities prevent further identification and quantification of the secondary N sources and its connection to ANR in altered MORB samples.

Veins from U1502B contain more abundant N (14–180 µg/g) with extremely low $\delta^{15}N$ values from −3.8‰ to −20.4‰ (Fig. 1; Supplementary Data 2). Because the vein minerals (particularly the $NH_4^+$-bearing minerals, e.g., albite, epidote, and chlorite) were precipitated at relatively high temperatures (>200 °C) that do not favor microbial activity, N contribution from microbial biomass to the samples can be excluded. The N concentrations of U1502B veins show good correlations with not only the modes of N-bearing secondary silicate

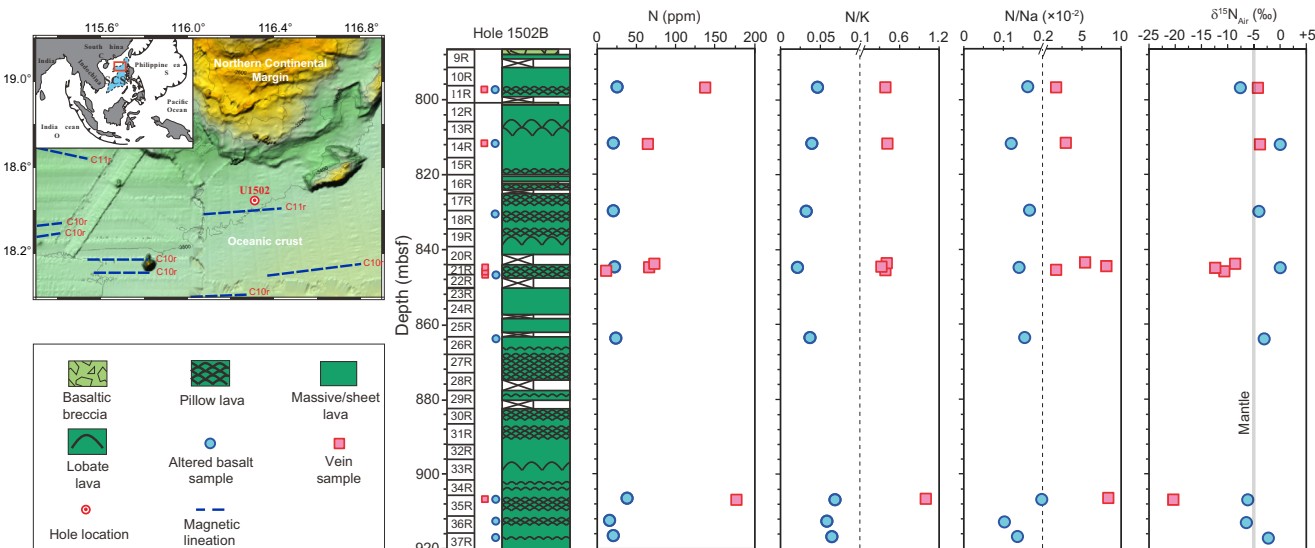

**Fig. 1 | IODP Hole U1502B location and downhole lithology, N concentration, molar N/K and N/Na ratios, and δ15N values of altered basalts and hydrothermal veins.** In the location map, dashed lines represent the magnetic

anomaly lineations. Note the scale change in the N/K and N/Na ratios marked by the dashed lines.

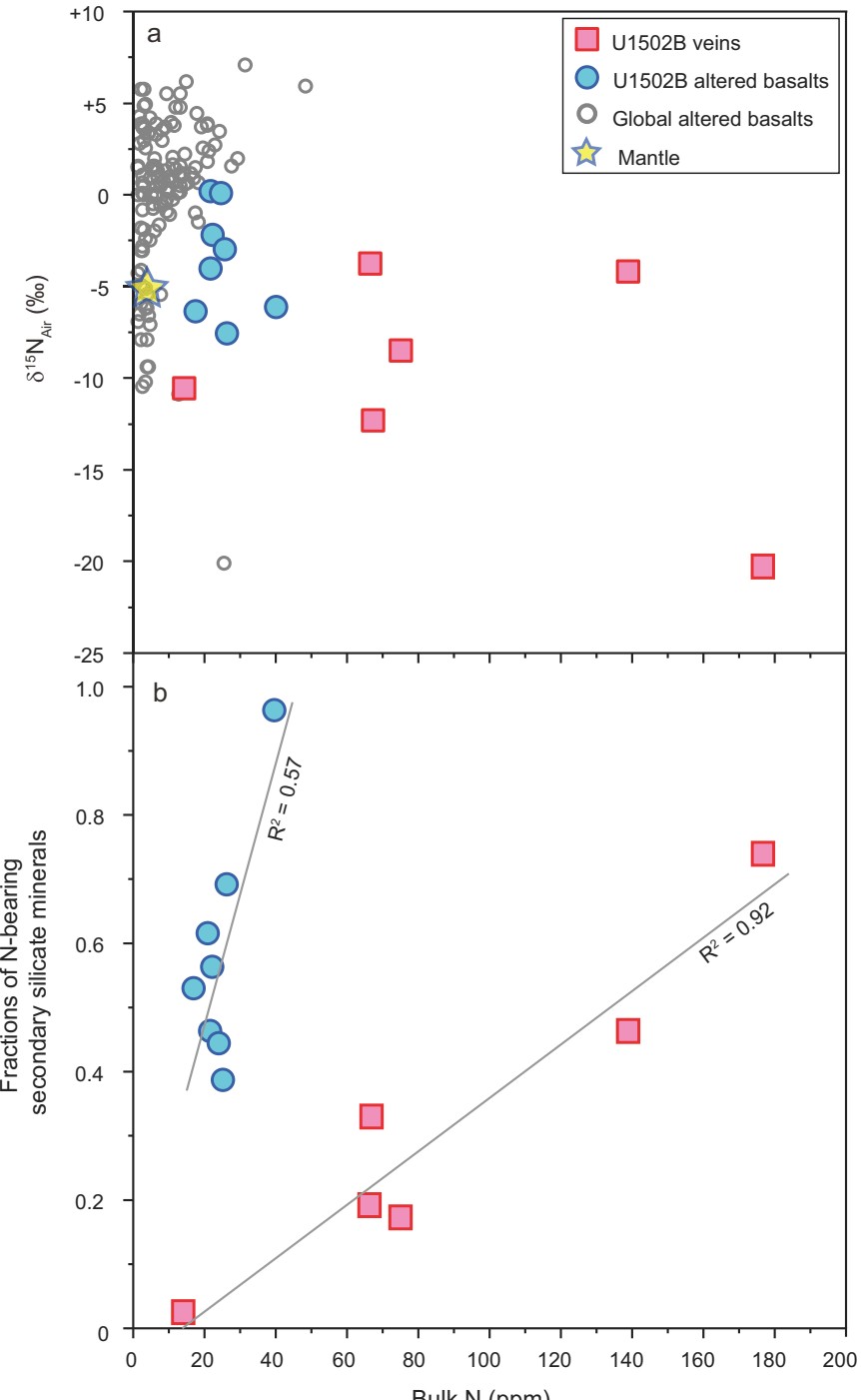

**Fig. 2 | Comparison of nitrogen concentrations with δ¹⁵N values and the fractions of nitrogen-bearing secondary silicate minerals in altered basalts and hydrothermal veins.** Data of global altered basalts in panel (**a**) are from Li and Li[29] and reference therein. Nitrogen-bearing secondary minerals in panel (**b**) include albite, chlorite, augite, and/or epidote.

minerals (Fig. 2), but also the concentrations of alkali elements Na and K and trace elements Rb and Cs (Fig. 2, Supplementary Fig. 5). These correlations verify that the N in the vein samples exists mainly in the form of $NH_4^+$ that substitutes $Na^+$ and/or $K^+$ in the vein minerals. The signatures of very low $K_2O$ concentrations (0.06 ± 0.03 wt.%; 1 SD; Supplementary Data 2) and high N/K molar ratios (0.51 ± 0.25; 1 SD), but high $Na_2O$ concentrations (0.55 ± 0.58 wt.%; 1 SD) and low Na/K molar ratios (0.05 ± 0.03; 1 SD) of the vein samples, which are similar to those of high-T altered oceanic crust (e.g., sheeted dikes and gabbros) in global oceans,

further suggest that the secondary $NH_4^+$ in the U1502B veins is mainly hosted in the $Na^+$ site rather than the $K^+$ site[30,40]. Using N/Na molar ratio to remove the modal heterogeneity of $NH_4^+$-bearing minerals across samples, the U1502B vein samples still have higher N/Na molar ratios than not only their hosting altered basalts (Fig. 1) but also the high-T altered gabbroic sections in global oceanic basement (N/Na = 0.0006–0.0022)[30]. This difference suggests that the $NH_4^+$ concentration was higher in the focused-flow hydrothermal fluids that formed the U1502B veins than in the diffusive fluids that dominantly altered the gabbros and basalts in global oceanic basements[30,40].

## Nitrogen sources

It is interesting to observe that the veins display a persistent uphole increase in $\delta^{15}N$ from −20.3‰ to −3.8‰ (Fig. 1) and a decrease in N/Na ratio from 0.084 to 0.018. Because the mineral assemblage and formation T of the veins did not change significantly throughout the core[36–38], these gradual changes in N/Na ratio and $\delta^{15}N$ value cannot be explained by temperature effect on elemental partitioning[34] and N isotope fractionation[35]. Neither these changes can be explained by low-T organic contamination which would otherwise give higher N/Na ratios for shallower samples. These extremely low $\delta^{15}N$ values can neither be attributed to a microbial source because the $\delta^{15}N$ values of living microbial biomass in hydrothermal systems mostly fall in a range of −4‰ to +7‰[41]. Instead, these uphole changes are best explained by a two-component mixing model (Fig. 3a; "Methods"). The shallow component is characterized by low N/Na ratios (or low $NH_4^+$ concentrations) and positive $\delta^{15}N$ values, consistent with a seawater-dominated fluid source containing $^{15}N$-enriched surface $NH_4^+$. The deep component is characterized by high N/Na ratios (or high $NH_4^+$ concentrations) and extreme $^{15}N$ depletions (−12‰ to −21‰). Taking the isotope fractionation factor between mineral (e.g., albite) and aqueous $NH_4^+$ (-2‰ to 3‰ in the range of 200–300 °C)[35] into consideration, the $\delta^{15}N$ value of $NH_4^+$ in the deep fluids is expected to vary from <−14‰ to −23‰.

Known N reservoirs that possibly contributed to the deep fluids, i.e., the mantle source of MORB (~ −5‰ for both N-MORB and E-MORB)[24,25], dissolved atmospheric $N_2$ (-0‰) and $NO_3^-$ (+3‰ to +8‰)[21] in seawater, and marine organic matter/sediments including dissolved organic N and $NH_4^+$ in interstitial water (mostly in the range of +2‰ to +10‰[22], up to 17‰[23]), are all much more $^{15}N$-enriched. Thus, the extremely low $\delta^{15}N$ values of $NH_4^+$ in the deep fluids have to be attributed to isotope fractionation associated with abiotic reactions in the deep hydrothermal system. In light of the N isotope fractionation factors determined by laboratory experiments[17,18] and theoretical calculations[35,42], only two abiotic processes can produce remarkably $^{15}N$-depleted $NH_4^+$ in submarine hydrothermal environments. One is partial reduction of $NO_3^-$. However, given that $NO_3^-$ in bottom seawater has a $\delta^{15}N$ value of about +5 ± 2‰[21], the lowest $\delta^{15}N$ value of $NH_4^+$ product can be −10‰ (at 200 °C) to −6‰ (at 300 °C)[18,35] (Fig. 3c), which are much less negative than the observed $\delta^{15}N$ values of $NH_4^+$ in U1502B deep fluids. The other is partial reduction of $N_2$. Although $NH_4^+$ is more enriched in $^{15}N$ than $N_2$ at isotopic equilibration[42], experimental studies reveal that $N_2$ is extremely difficult to reach isotope equilibration with other N species due to its strong triple bond as an energy barrier for isotope exchange even at high temperatures of 300 – 800 °C[43,44]. As a result, abiotic reactions involving $N_2$ generally produce large kinetic isotope fractionations, e.g., −18‰ to −16‰ at 600–800 °C during $NH_3$ decomposition[43], a reverse process of ANR. From these values and the equilibrium isotope fractionation factors[42], kinetic isotope enrichment factors of −13‰ to −16‰ can be deduced for ANR at 600–800 °C ("Methods"). Accordingly, if the initial $N_2$ is from the upper mantle source (−5‰), ANR-produced $NH_4^+$ at 600–800 °C can have $\delta^{15}N$ values as low as −21‰ (Fig. 3c). If ANR occurs at lower temperatures, which is highly likely for a submarine hydrothermal system, the magnitude of kinetic isotope fractionation should be larger (although not quantitatively constrained)[43] and thus result in more negative $\delta^{15}N$ values in the $NH_4^+$ product (Fig. 3c). $N_2$ is commonly exsolved during mantle upwelling and partial melting beneath mid-ocean ridges[24,25]. The $N_2$ outflux at global mid-ocean ridges ($1.6 \times 10^{10}$ mol·yr$^{-1}$)[45] is at the same order of magnitude with the $N_2$ outflux at global arc volcanoes ($2.0–6.7 \times 10^{10}$ mol·yr$^{-1}$)[46], and thus can provide a sustainable source for ANR. Therefore, the extremely $^{15}N$-depleted $NH_4^+$ in the deep fluids, as recorded by the vein minerals, can be best explained by abiotic reduction of mantle $N_2$ in the deep oceanic basement through the interaction between deep fluids and $Fe^{2+}$-bearing minerals (Fig. 4). The relatively large $\delta^{15}N$ range of the deep-fluid endmember in Fig. 3a, b

may be attributed to different extents of ANR (Fig. 3c) and/or heterogeneous mixing with surface $NH_4^+$.

Given that the hydrothermal system below U1502B was sustained by heat flux from intrusive mafic dykes[38], the most likely $Fe^{2+}$-bearing minerals for ANR are pyroxene (ferrosilite as the $Fe^{2+}$-rich endmember) and possibly olivine (fayalite as the $Fe^{2+}$-rich endmember) in these rocks. The ANR can be expressed by Reactions (1) − (2) for ferrosilite or Reactions (3) − (4) for fayalite[22]:

$$9FeSiO_3 + N_2 + 3H_2O + 2H^+ \rightarrow 3Fe_3O_4 + 9SiO_2 + 2NH_4^+ \tag{1}$$

$$6FeSiO_3 + N_2 + 3H_2O + 2H^+ \rightarrow 3Fe_2O_3 + 6SiO_2 + 2NH_4^+ \tag{2}$$

$$9Fe_2SiO_4 + 2N_2 + 6H_2O + 4H^+ \rightarrow 6Fe_3O_4 + 9SiO_2 + 4NH_4^+ \tag{3}$$

$$3Fe_2SiO_4 + N_2 + 3H_2O + 2H^+ \rightarrow 3Fe_2O_3 + 3SiO_2 + 2NH_4^+ \tag{4}$$

The extreme $^{15}N$ depletion in the vein samples progressively diminishes toward the surface, associated with a decrease in N/Na ratio (Fig. 1). This indicates the ANR-produced $NH_4^+$ in the deep fluid below U1502B has been progressively overprinted by surface $NH_4^+$ following enhanced mixing with seawater-derived fluids towards seafloor (Fig. 4). This mixing effect can also explain why the ANR signal is rarely recorded in global altered oceanic basalts, because the alteration fluids in the highly porous upper section of oceanic crust are dominated by relatively less evolved seawater that contains sediment-derived $NH_4^+$[29]. In contrast, in more evolved deep fluids, $^{15}N$-enriched surface $NH_4^+$ from seawater-dominated fluid could have been stripped away by oceanic basement rocks along the pathway that the shallow fluid was pumped down. Thus, the deep fluid can preserve the $^{15}N$-depleted signature of ANR, as manifested by the minerals deposited from the focused flow channel of these deep fluids. This mixing discrepancy is clearly shown by the difference in N concentration and isotope composition between veins and their direct wall-rock basalts in Hole U1502B (Figs. 1–2).

Nevertheless, the $\delta^{15}N$ values of U1502B altered basalts are clearly lower than the range of surface $NH_4^+$ (+2‰ to +10‰) and extend towards the negative deep fluid values (Fig. 3b). This implies that the ANR signal could still have been partially implanted into altered basalts, but significantly weakened due to surface $NH_4^+$ overprinting. Significant $^{15}N$ depletion (as low as −20‰) has also been observed in the hydrothermally altered oceanic basalts from the western Pacific (ODP Sites 1149 and 801 with ages of 130–170 Ma and fast half-spreading rates of 51 − 80 mm·yr$^{-1}$)[27] and the mid-Atlantic (DSDP Site 417 with an age of 120 Ma and a slow half-spreading rate of -12.5 mm·yr$^{-1}$)[29], as well as their blueschist- to eclogite-facies metamorphic equivalents[22,47]. This may imply that ANR has occurred more commonly in worldwide submarine hydrothermal systems.

## Implications to Earth's N cycling

To assess the ANR contribution to the seawater $NH_4^+$ reservoir, we employed a commonly used model ("Methods") to estimate the $NH_4^+$ concentration of the deep fluid, which gave a ballpark range of 12.5–15.0 mM. These values are higher than the few available $NH_4^+$ concentration data of high-T (300–380 °C) hydrothermal fluids without sedimentary N contribution (e.g., <0.01 mM; Supplementary Data 4) but close to the $NH_4^+$ concentrations of the high-T (300–315 °C) hydrothermal fluids in the Guaymas Basin (12.9–15.2 mM), which were speculated to originate from thermal degradation of organic matter[48]. Thus, the high $NH_4^+$ concentration of the deep fluid at U1502B may represent an upper end of ANR contribution to hydrothermal vents at a global scale.

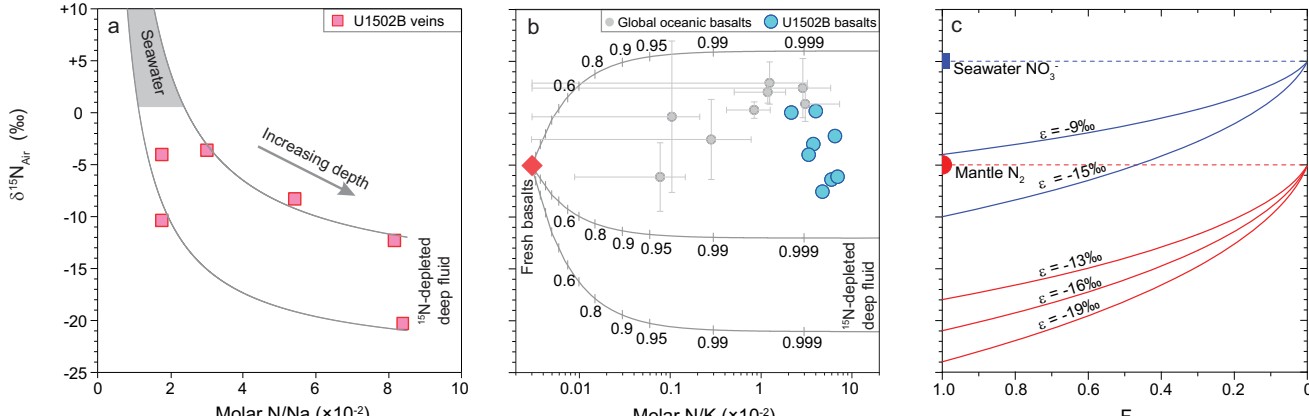

**Fig. 3 | Nitrogen isotope modeling. a, b** Two-endmember mixing model for hydrothermal veins and altered basalts, respectively. **c** The isotopic values of potential endmember $NH_4^+$ from partial reduction of $N_2$ (red curves) or $NO_3^-$ (blue curves) based on Rayleigh distillation model. In panel a: the mixing curves along the upper and lower data boundaries of vein sample data converge at a confined shallow endmember with positive $\delta^{15}N$ values and low N/Na ratios, which are consistent with seawater. The lower endmember displays variable $\delta^{15}N$ values from −12‰ to −21‰, likely due to the variable extent of ANR (see text). In panel b: reference mixing curves illustrate addition of secondary $NH_4^+$ into fresh basalts from hydrothermal fluids with $\delta^{15}N$ values of +6‰ (a typical seawater value), −12‰, and −21‰. Ticks on the mixing curve mark the proportions of secondary N in total N. Note the U1502B data sit far away from the fresh basalt endmember and suggest

that >99% of the N originated from secondary sources involving both seawater and $^{15}N$-depleted hydrothermal fluids. For comparison, DSDP/ODP/IODP basalts from global oceans are also plotted, with each point and associated error bars representing the average and 1 SD of the data from an individual site (see ref. 29 for the full dataset). In panel (**c**): F refers to the fraction of remaining N after the reaction, from no reduction (F = 1) to complete reduction (F = 0). See text for the determination of the isotope enrichment factors (ε) for the $NH_4^+ - NO_3^-$ pair and the $NH_4^+ - N_2$ pair. The results illustrate that, depending on the extent of the reaction, the $\delta^{15}N$ value of the accumulated $NH_4^+$ product may increase from extremely negative values at low degree of $N_2$ reduction to close to its source value at high degrees of reduction.

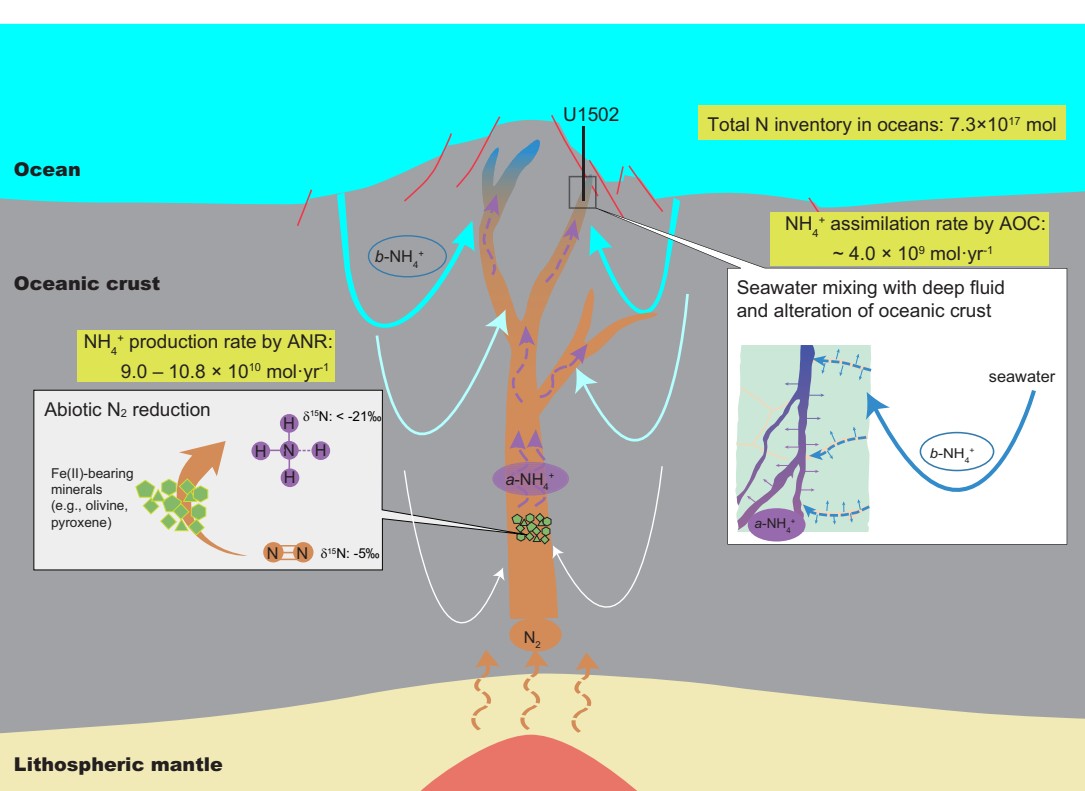

**Fig. 4 | Schematic diagram (not to scale) showing abiotic $N_2$ reduction in deep fluids, mixing of seawater into deep fluids, and alteration of oceanic crust in mid-ocean ridge hydrothermal systems.** $a$-$NH_4^+$ denotes the $NH_4^+$ produced from abiotic $N_2$ reduction by $Fe^{2+}$-bearing minerals; $b$-$NH_4^+$ denotes biogenic $NH_4^+$ in seawater. Along the circulation pathway of deep seawater, $b$-$NH_4^+$ was

progressively consumed via $NH_4^+$ assimilation into secondary silicate minerals formed during seawater-rock interaction. Thus, the impact of $b$-$NH_4^+$ on the N signature of hydrothermal fluids was less in depths but more and more prominent toward the seafloor.

Applying this concentration range and the flux ($0.72 \times 10^{13}$ kg·yr$^{-1}$) of deep high-T hydrothermal fluids[49], an upper limit of $NH_4^+$ flux of $9.0$–$10.8 \times 10^{10}$ mol·yr$^{-1}$ from high-T fluids into seawater can be obtained ("Methods"). The lower limit of ANR-generated $NH_4^+$ flux could be much smaller, but it cannot be quantified yet due to the lack of data. Nevertheless, this estimate provides a reference for a first-order understanding of the relative role of ANR in the geological N cycle.

The estimated $NH_4^+$ flux here is comparable to the outflux of $N_2$ in mid-ocean ridges ($1.6 \times 10^{10}$ mol·yr$^{-1}$)[45] and arc volcanoes ($2.0$–$6.7 \times 10^{10}$ mol·yr$^{-1}$)[46], implying that ANR could have been an important mechanism to retain about half or more of the degassed mantle $N_2$ into the oceans. The ANR-produced $NH_4^+$ can barely impact the modern marine N cycle given the large N reservoir ($7.3 \times 10^{17}$ mol) in global oceans[50]. However, ANR could have been a key process to supply $NH_4^+$ to Earth's early oceans. Early Earth's surface contained more (ultra) mafic components with higher abundance of $Fe^{2+}$-bearing minerals such as olivine and pyroxene which could have facilitated more intensive serpentinization (a hydrothermal alteration process of olivine and pyroxene) than in the Proterozoic-Phanerozoic oceans[51]. Since ANR can be closely associated with serpentinization[16], ANR could be more productive in the Hadean and Archean oceans. $NH_4^+$ extracted from fluid inclusions in 3.5 Ga hydrothermal minerals show strong $^{15}N$ depletion with $\delta^{15}N$ values as low as −13‰[52], which might imply an $NH_4^+$ source dominantly derived from ANR in the Archean submarine hydrothermal systems, although the ANR signal could be obscured in the altered basalt record by more rigorous biological N recycling in the Archean oceans[53]. Based on above estimated $NH_4^+$ production rate from ANR and the volume of Archean oceans ($1.3$–$2.3$ times of modern ocean volume[54]), the $NH_4^+$ concentration of the early oceans could reach the modern seawater N level ($NO_3^-$, $NO_2^-$ and $NH_4^+$ all combined) within 0.24 million years, if no significant $NH_4^+$ sink is considered. If assume the Archean oceanic basement could assimilate $NH_4^+$ from seawater comparably to the modern oceanic basement ($4.0 \times 10^9$ mol·yr$^{-1}$; "Methods"), it would have taken 0.25 million years for the early oceans to accumulate $NH_4^+$ to the modern seawater N level. This suggests that $NH_4^+$ level could have been very quickly built up in the prebiotic ocean for the development of habitability and the origin of life, as well as supporting a microbial ecosystem (see Supplementary Information).

## Methods

### Sample Preparation

Altered basalt and vein samples were cleaned by removing surface materials and crushed into small size (1–2 mm) before further washed by deionized water. After drying at 50 °C in an oven, the grains were ground to <200 mesh by hand in an agate mortar for further analyses.

### Mineralogy

Mineralogical analysis of altered basalts and veins were performed on a D8 ADVANCE X-ray diffractometer (XRD) with a copper Kα tube at the State Key Laboratory of Tropical Oceanography, South China Sea Institute of Oceanology, Chinese Academy of Sciences. Instrument settings were 40 mA, 40 kV, angle steps of 0.02°, a counting time per step of 0.15 s and scanning angles of 5–90°. The quantitative compositions of minerals were calculated using the K value method[55] with DIFFRAC.EVA 5.1.0.5 software. The intensity ratio on the XRD curve was obtained from PDF2–2004. The NIST corundum standard SRM 1976 was used for the correction of instrumental goniometers. The analytical error is 5% – 10%.

### Major and trace elements

Major and trace elements of altered basalts were determined at the Wuhan SampleSolution Analytical Technology Co., Ltd. (Wuhan, China). Major elements were measured using a Rigaku Primus II X-ray

Fluorescence spectrometer (XRF). Altered basalt powders were fluxed with 6.0 g co-solvent ($Li_2B_4O_7$: $LiBO_2$: $LiF = 9$:$2$:$1$) and 0.3 g oxidant ($NH_4NO_3$) at 1150 °C to make homogeneous glass discs for the XRF analysis. The standards GBW07103, GBW07111, and GBW07114 were used for data calibration. Analytical error is better than 5% of the absolute concentrations. Trace elements of altered basalts were analyzed using an Agilent 7700e inductively coupled plasma mass spectrometer (ICP-MS) after acid digestion of samples in high-pressure Teflon vessels. The USGS reference standards BHVO, BCR-2, and RGM-2 were used for data calibration. The analytical error is better than 10% of the absolute concentrations.

Major and trace elements of veins were analyzed in the Laboratory of Ocean Lithosphere and Mantle Dynamics, the Institute of Oceanology, Chinese Academy of Sciences. Major elements were analyzed using an Agilent 5100 inductively coupled plasma optical emission spectrometer (ICP-OES). About 50 mg of sample powders and 250 mg of lithium metaborate ($LiBO_2$) were mixed in a platinum crucible and melted in a muffle furnace at 1050 °C for 1 h. The melt was further heated and spun on a Bunsen burner at about 1000 °C to ensure that all melts formed a coherent single drop, which was finally poured into 5% $HNO_3$ for dissolution. Repeated analyses of USGS reference standards (BHVO-2 and BCR-2) gave an analytical error better than 5% of the absolute concentrations. Trace elements of veins were analyzed using an Agilent 7900 ICP-MS. About 50 mg of sample powders were dissolved in an acid mixture of HCl, $HNO_3$, and HF in a high-pressure jacket equipped Teflon beaker for 15 h, and then re-dissolved with 20% $HNO_3$ for 2 h until complete digestion. The reference materials BHVO-2, GSP-2 and W-2 were used for data calibration. The analytical error is better than 10% of the absolute concentrations.

### Nitrogen concentration and isotope composition

Nitrogen concentration and isotope composition of altered basalts and veins were analyzed by the offline sealed-tube combustion and extraction methods coupled with continuous flow isotope ratio mass spectrometry[56] at the Stable Isotope Geochemistry Laboratory, University of Alberta. About 100 mg sample powders, 400 mg CuO reagents and pre-cleaned (1200 °C for 3 h) quartz wool were sequentially loaded into a pre-cleaned (1200 °C for 3 h) one-end sealed quartz tube. The sample tube was attached on a custom-made metal manifold to pump overnight and then sealed under high vacuum. The entire sealed sample tube was put into a programmable muffle furnace to combust at 900 °C for 8 h followed by 600 °C for 2 h. The combusted tube was then loaded into a tube cracker attached on the metal manifold and cracked under high vacuum. The released $N_2$ was purified using a liquid $N_2$ trap and quantified by a capacitance manometer. Finally, the $N_2$ gas was introduced by an ultrahigh-purity helium stream through a GasBench II interface into a Thermo Finnegan MAT 253 isotope ratio mass spectrometer for N isotopic measurement. All N isotope data were reported in the δ notation, which is defined as $\delta^{15}N_{sample} = (^{15}N/^{14}N)_{sample}/(^{15}N/^{14}N)_{standard} - 1$, where the standard is atmospheric $N_2$. Repeated analyses of certified reference material (the low-organic content soil standard and high-organic content sediment standard from Elemental Microanalysis) and samples gave an analytical error better than 6% (2 SD) of the absolute value for N concentration and 0.2‰ (2 SD) for $\delta^{15}N$ value.

### Data modeling

#### Nitrogen mixing modeling

**(a) Hydrothermal veins.** Since the vein minerals assimilated $NH_4^+$ directly from the hydrothermal fluid as they precipitated, the uphole gradual decrease in N/Na ratio and increase in $\delta^{15}N$ value are therefore considered to reflect decreasing $NH_4^+$ concentration with increasing $\delta^{15}N$ value of the hydrothermal fluids as they migrated upwards. This can be best attributed to gradually intensified mixing of another fluid (with a low N/Na ratio and a high $\delta^{15}N$ value) from the top into the up-

flowing deep fluid. For quantitative modeling of this case scenario, N concentration variations caused by inter-sample modal heterogeneity of $NH_4^+$-bearing minerals have to be removed first. For this purpose, N/Na molar ratio (rather than N concentration) is used together with $\delta^{15}N$ for the mixing modeling. Accordingly, this two-endmember mixing process can be described as:

$$\left(\frac{N}{Na}\right)_{Sample} = \left(\frac{N}{Na}\right)_{TF} \times f + \left(\frac{N}{Na}\right)_{DF} \times (1-f) \quad (5)$$

$$\delta^{15}N_{Sample} = \frac{\left(\frac{N}{Na}\right)_{TF} \times f \times \delta^{15}N_{TF} + \left(\frac{N}{Na}\right)_{DF} \times (1-f) \times \delta^{15}N_{DF}}{\left(\frac{N}{Na}\right)_{TF} \times f + \left(\frac{N}{Na}\right)_{DF} \times (1-f)} \quad (6)$$

in which *TF* denotes top fluid, *DF* denotes deep fluid, and f refers to the fraction of the top fluid in any hydrothermal fluid. It should be noted that, because the partition coefficient of $NH_4^+$ and $Na^+$ between hydrothermal fluid and the multiple $Na^+$-bearing secondary minerals (albite, epidote, chlorite) are not all available, these factors as well as the N isotope fractionation factors between the hydrothermal fluids and minerals (which are relatively small[35]), were not taken into consideration for simplification. Therefore, $\left(\frac{N}{Na}\right)_{TF}$, $\left(\frac{N}{Na}\right)_{DF}$, $\delta^{15}N_{TF}$ and $\delta^{15}N_{DF}$ do not represent the $\frac{N}{Na}$ ratios and the $\delta^{15}N$ value of the top and deep fluids but the values of secondary minerals precipitated from the top and deep fluids, respectively.

The modeling results (Fig. 3a) indicate that the lower endmember has a relatively large $\delta^{15}N$ range with high N/Na ratios, e.g., if using a N/Na ratio of 0.085 (close to the largest value of the veins), a $\delta^{15}N$ range of −12‰ to −21‰ is required to cover the data. In contrast, the upper endmember is characterized by much more narrow ranges with low N/Na ratios and high $\delta^{15}N$ values, which are consistent with a seawater source.

**(b) Altered basalts**. Different to hydrothermal veins whose N was solely sourced from hydrothermal fluid, altered oceanic basalts contain a small amount of mantle-inherited N (up to 2 μg/g, $\delta^{15}N = -5‰$)[24,25,27,29–31]. Further addition of $NH_4^+$ into oceanic basalts can be induced by hydrothermal alteration mostly occurring at near seafloor with relatively low temperatures, in which $K^+$-rich secondary minerals commonly precipitate and simultaneously assimilate $NH_4^+$ from the ambient fluid[26–31]. In this case scenario, because the $NH_4^+$ in hydrothermal fluid can be taken as an unlimited reservoir in relative to the inherited $NH_4^+$ in fresh basalts due to the high water/rock ratio near seafloor, the mixing model for addition of hydrothermal $NH_4^+$ into altered basalts is slightly different to the fluid mixing model above for hydrothermal veins, but instead can be described as[27,29–31]:

$$\delta^{15}N_{Sample} = \frac{\left(\frac{N}{K}\right)_{FB} \times \delta^{15}N_{FB} + \left[\left(\frac{N}{K}\right)_{Sample} - \left(\frac{N}{K}\right)_{FB}\right] \times \delta^{15}N_{DF}}{\left(\frac{N}{K}\right)_{Sample}} \quad (7)$$

in which *FB* denotes fresh basalts. N/K molar ratio is used to remove the effect of modal heterogeneity of $K^+$-bearing minerals on N concentrations of altered basalts. The N/K ratio of fresh basalt is calculated from the N concentration[29] and K concentration[57] of fresh oceanic basalts. Again, for simplification, the partition coefficients of $NH_4^+$ and $K^+$ and N isotope fractionation factors between hydrothermal fluid and the multiple $K^+$-bearing secondary minerals were not taken into consideration. Therefore, $\left(\frac{N}{K}\right)_{DF}$ and $\delta^{15}N_{DF}$ do not represent the $\frac{N}{K}$ ratio and the $\delta^{15}N$ value of deep fluids but the values of secondary minerals precipitated from deep fluids.

In Fig. 3b, mixing curves were illustrated for $NH_4^+$ addition from seawater ($\delta^{15}N = 6‰$, which is an average value of marine organic matter and sediments)[22] and two variably $^{15}N$-depleted deep fluids ($\delta^{15}N = -12‰$ and −21‰, respectively). The results show that the N in

U1502B basalts is dominated by secondary N from a mixture of seawater and $^{15}N$-depleted deep fluids.

**N isotope fractionation during ANR**. The N isotopic effects associated with abiotic transformation between $N_2$ and $NH_3$ can be described as:

$$N_2 + 3H_2 \underset{k_2}{\overset{k_1}{\rightleftharpoons}} 2NH_3 \quad (8)$$

in which $k_1$ denotes the kinetic isotopic fractionation factor for abiotic $N_2$ reduction and $k_2$ denotes the kinetic isotopic fractionation for abiotic $NH_3$ decomposition. Consequently, the equilibrium isotope fractionation factor between $N_2$ and $NH_3$ ($\alpha_{N_2-NH_3}$) can be described as:

$$\alpha_{N_2-NH_3} = k_1/k_2 \quad (9)$$

Currently, the $k_2$ values have been only determined for the temperature range of 600–800 °C (0.9841 to 0.9823)[43]. Integrated with the $\alpha_{N_2-NH_3}$ values in the same temperature range (1.0029–1.0017)[42], the $k_1$ values are calculated to be 0.9869 to 0.9845, which are equivalent to isotope enrichment factors of −16‰ to −13‰ for the temperature range of 600–800 °C. The isotope fractionation factors at lower temperatures cannot be quantified at the moment due to the lack of $k_2$ values at lower temperatures, but are expected to be larger at lower temperatures[43], which would fit our data even better.

**$\delta^{15}N$ of accumulated $NH_4^+$ from ANR**. Because of the strong triple bond of $N_2$, the $NH_4^+$ product will not exchange N isotope compositions with remaining $N_2$ after ANR. Therefore, the $\delta^{15}N$ value of accumulated $NH_4^+$ product from a certain degree of ANR follow the traditional Rayleigh model, which can be described by the equation below[43,58]:

$$\delta^{15}N_p = \delta^{15}N_0 - \varepsilon \cdot \frac{f \cdot lnf}{1-f} \quad (10)$$

in which $\delta^{15}N_O$ and $\delta^{15}N_p$ refer to the N isotopic value of initial N species (+6‰ for marine $NO_3^-$ or −5‰ for mantle $N_2$) and the product ($NH_4^+$), respectively; $\varepsilon$ denotes the isotopic enrichment factor between the product and initial N species; $f$ denotes the fraction of remaining N after the reaction. Examples of the modeling are illustrated in Fig. 3c.

**Ammonium productivity from ANR**. The common method to derive the $NH_4^+$ concentration of deep hydrothermal fluids from the $NH_4^+$ concentrations of vein samples is to use the $NH_4^+$ partition coefficient between the vein sample and fluid, which can be described by the equation below:

$$C_{NH_4^+}^{fluid} = C_{NH_4^+}^{vein} / D^{vein-fluid} \quad (11)$$

in which $C$ is $NH_4^+$ concentration of a vein sample or fluid; $D^{vein-fluid}$ is the total partition coefficient of $NH_4^+$ between vein and fluid, which can be calculated by the following equation:

$$D^{vein-fluid} = \sum n \cdot K_d^{mineral-fluid} \quad (12)$$

in which $n$ is the fraction of individual $NH_4^+$-bearing minerals (albite, epidote and chlorite with augite in some samples in this study); $K_d^{mineral-fluid}$ is the partition coefficient between a $NH_4^+$-bearing vein mineral and fluid.

While the partition coefficients of $NH_4^+$ between these silicate minerals and fluid have not been all determined, they can be calculated by the Blundy and Wood model[59], which suggests a non-linear least-squares regression relationship between the partition coefficients and the radii of isovalent cations ($Li^+$, $Na^+$, $K^+$, $NH_4^+$, $Rb^+$, $Cs^+$)[60–62] (Supplementary Fig. 7). The calculation results (Supplementary Data 3) show that the mineral-fluid partition coefficients of $NH_4^+$ and $Rb^+$ are almost

identical, because of their similarity in cation radius. This conclusion of similar partitioning behavior of $NH_4^+$ and $Rb^+$ in mineral-fluid system has been not only demonstrated by a previous study for phengite[63], but also supported by the good correlation between the concentrations of $NH_4^+$ and $Rb^+$ in the studied vein samples ($R^2 = 0.80$; Supplementary Fig. 5).

Because the $D^{vein-fluid}$ values of $NH_4^+$ and $Rb^+$ are almost identical, the $NH_4^+$ concentrations of fluid can be simply calculated from the equation below:

$$C_{NH_4^+}^{fluid} = C_{NH_4^+}^{vein} \times \frac{C_{Rb}^{fluid}}{C_{Rb}^{vein}} \qquad (13)$$

The advantage of Eq. (13) is that most of the uncertainties associated with Eqs. (11, 12), including (i) $K_d$ determination for individual minerals by laboratory experiments and the interacting effect with other cations (e.g., $Na^+$) during $NH_4^+$ partitioning[34], and (ii) the fraction of each $NH_4^+$-bearing mineral in bulk rock, can be canceled out. It should be noted that Eq. (13) cannot be applied to altered basalts because the initial N and Rb in the basalts prior to alteration are decoupled from hydrothermal fluids.

This method using Eq. (13) requires to know the $Rb^+$ concentration of fluid, which is difficult to constrain due to the lack of well-preserved fluid samples. Here, we use the concentrations of high-temperature (> 300 °C) hydrothermal fluids in modern submarine hydrothermal systems as a reference. To obtain representative aqueous elemental data of deep fluids, we only used the fluid data in literature with $[Mg^{2+}] = 0$, which indicates the correction for seawater contamination. The filtered data (Supplementary Data 4) gave very consistent concentrations of univalent cations (e.g., $Na^+$, $K^+$, and $Rb^+$) although the high-temperature fluids came from a variety of different submarine localities around the world[64–69].

We use the samples with the lowest $\delta^{15}N$ values in two veins (a silicate vein with high quartz content and a silicate vein with low quartz content; N = 67.4–180.4 μg/g; Rb = 0.84–2.72 μg/g; Supplementary Data 2) to best represent the deep fluid with least seawater contamination (Rb = 31 mM; Supplementary Data 4), from which $NH_4^+$ concentrations of 12.5–15.0 mM can be deduced for the deep fluids at U1502B. Applying this concentration range and the flux of deep high-T hydrothermal fluids ($0.72 \times 10^{13}$ kg·yr$^{-1}$)[49], an upper limit of $NH_4^+$ flux of $9.0–10.8 \times 10^{10}$ mol·yr$^{-1}$ from high-T fluids into seawater can be obtained.

**$NH_4^+$ uptake by global altered basalts.** Ammonium in seawater can be assimilated into oceanic basement rocks during seafloor alteration/weathering[26–31]. The $NH_4^+$ assimilation flux in modern oceans can be calculated using the following equation:

$$F = C \times D \times P \qquad (14)$$

in which $F$ is $NH_4^+$ assimilation flux, $C$ is the N concentration of altered basalts (global average: 9.9 μg/g)[29], $D$ is density of altered basalts (2.7 g·cm$^{-3}$), $P$ is oceanic basalt production rate (2.1 km$^3$·yr$^{-1}$)[70]. This gave an $NH_4^+$ assimilation flux of $4.0 \times 10^9$ mol·yr$^{-1}$ in modern global oceans.

It should be noted that, the $NH_4^+$ assimilation rate of altered basalts is strongly dependent on the $NH_4^+$ availability in local environment (e.g., absence or occurrence of N-rich organics and sediments)[29]. Thus, the $NH_4^+$ assimilation flux in the early oceans was likely not as large as that in modern oceans.

## Data availability

The data generated in this study have been deposited in Mendeley Data at https://doi.org/10.17632/xgwv73pj7b.2.

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

## Acknowledgements

We thank the dedicated efforts of the crew and scientific staff of the Drillship JOIDES Resolution during IODP Expeditions 367 and 368. We thank L. Cao and Z. Zhang for XRD analyses. This work was financially supported by the Key Project of Joint Geological Fund of National Natural Science Foundation of China (U2244221), the Major Research Plan on West-Pacific Earth System Multispheric Interactions (92158205), the Guangdong Special Support talent team Program (2019BT02H594), the National Key R&D Program of China (2021YFC3100600), the National Natural Science Foundation of China (42372080) and the Ministry of Science and Technology of China (2023YFF0803402) to Z.S., the National Natural Science Foundation of China (42406067) to L.S., and the NSERC-Discovery Grant to L.L.

## Author contributions

L.S. designed the experiments, analyzed the data and prepared the manuscript. K.L. and Y.Z. contributed to data collection and interpretation. Z.S. and L.L. conceptualized and supervised the project, analyzed the data and revised the manuscripts.

## Competing interests

The authors declare no competing interests.
