## [Peer Review file · Nature Communications]

Abiotic N₂ reduction in submarine hydrothermal systems could quickly fertilize prebiotic oceans

Corresponding Author: Dr Long Li

Version 0:

Reviewer comments:

Reviewer #1

(Remarks to the Author)

Review of: Abiotic nitrogen reduction in submarine hydrothermal system

By: Sun et al.

Recommendation: Reconsider after revision.

Manuscript summary: The authors report on nitrogen isotopic variation in NH₄⁺ in samples from an Oligocene mid-ocean ridge hydrothermal system and conclude that the extremely 15N-depleted NH₄⁺ observed in deep fluids, as recorded by vein minerals, can be best explained by abiotic reduction of mantle N₂ in the deep oceanic basement through the interaction between the deep fluids and Fe(II)-bearing minerals (e.g., olivine, pyroxene). Upscaling this process, they argue that this ammonium flux is small relative to the large nitrogen inventory in modern oceans, but could be an important contributor to quickly fertilize the early oceans and supply NH₃ to the atmosphere on the prebiotic Earth.

Review summary: Prior to the evolution of nitrogenases, there was essentially no direct pathway known to derive ammonium from N₂ (assuming the early Earth atmosphere was ammonia-poor). The abiotic reduction of N₂ through hydrothermal basalt has been hypothesized but direct evidence has been lacking. The study by Sun et al. delivers this evidence and therefore adds an important piece to our picture of the early nitrogen cycle.

I find the logical flow of the investigation sound, the selected methods are robust, and the extrapolation of the findings are adequate. There are two major shortcomings that should be remedied before this manuscript could be reconsidered:

1) The writing has to be improved. In my specific comments I started out with language and grammar corrections but they became too extensive. Most notably, the last paragraph of the introduction reads like a Methods section. That last paragraph should define the question to be answered as well as tell the objective of the study which is not clear at all after reading the introduction. A more general reader might get highly confused. Moreover, the figures, especially the schematic, should be polished up.

2) The mechanism and catalytic reaction of N₂ reduction to ammonium by Fe(II)-bearing minerals has to be described more precisely. The authors should think more about the molecular mechanism that drives the reaction and suggest, if not identify, the Fe mineral serving as catalyst. Write out a proposed chemical reaction in a formula. The samples used are derived from an Oligocene bedrock, yet the authors draw conclusions to processes on the prebiotic Earth, well over 3 billion years ago. If the Fe minerals involved are better constrained, it can be argued that these minerals were also present in similar rock formations on the prebiotic Earth – which would essentially strengthen the extrapolation in time.

Given the quality of the data and the logic that the authors applied to draw conclusions, I would recommend reconsidering this work for publication in Nature Communications after these two main points and the following minor points were reworked.

Specific comments:

Title: The wording of the title is awkward. The title should be better fleshed out and point towards the significance of the

paper. For example: “Evidence for hydrothermal N₂ reduction to ammonium on the prebiotic Earth”.

L14, Abstract: “.. an essential component of the abiotic synthesis...”

L15, Abstract: take out “arguably” – such fill words are unnecessary.

L20, Abstract: name the nitrogen isotopes.

Abstract, overall: I suggest including this key finding in the abstract. “Therefore, the extremely ¹⁵N-depleted NH₄⁺ in the deep fluids, as recorded by the vein minerals, can be best explained by abiotic reduction of mantle N₂ in the deep oceanic basement through the interaction between the deep fluids and Fe(II)-bearing minerals (e.g., olivine, pyroxene).”

L30-36. Split this very long sentence into two. Suggestion: “...laboratory experiments. These demonstrated that..”

L33: “early Earth surface”

L34ff.: References 7-9 on abiotic nitrogen reduction reactions on the early Earth are important but outdated. Please include more recent work:

Nishizawa, M. et al. *Minerals* 11, 321 (2021)

Buessecker, S. et al. *Nat. Geosci.* <https://doi.org/10.1038/s41561-022-01089-9> (2022)

L40: “detecting an ANR signal...”

L51-57: Consider moving this to the methods and definitely modify the end of the introduction (see major comments).

L60-63: Clarify what N pool you are talking about – bulk, inorganic, organic?

L99: use plural, “basements”

L102, 103, and Fig. 1: Are these data (one data point) based on technical replicates? How was the heterogeneity in samples of the same core depth evaluated? The robustness especially of the measurement reporting –20.3‰ is important because it essentially creates the decreasing trend with depth. The authors should make sure this is not an outlier and a data point set of 3 such as for sample 21R would create more confidence.

L117-143: I like the logical flow of this paragraph very much. I think the way the conclusion that ¹⁵N depletion in ammonium originates from reduction of mantle N₂ is derived is quite reasonable and sound.

L134: Looking through the text, it is not explained where N₂ in the upper mantle actually comes from. The question of the ultimate source of N₂ would be important to evaluate how sustainable this reduced N source to the early oceans could have been. Please go into more detail (here or in the introduction) how N₂ in the upper mantle is thought to be derived and recycled.

L158: Provide very briefly more insight into the locations and formation history where else significant ¹⁵N depletion was reported from. This will demonstrate better that the proposed process could be a global phenomenon.

L173: I may have missed it somewhere in the supplements, but what is the basis of this calculated value. I suggest adding a reference to the supplements.

L183: How did serpentinization rate influence the GOE – please corroborate for readers not familiar with:

Leong, J.A.M., Ely, T. & Shock, E.L. Decreasing extents of Archean serpentinization contributed to the rise of an oxidized atmosphere. *Nat Commun* 12, 7341 (2021). <https://doi.org/10.1038/s41467-021-27589-7>

L191: Could an early biosphere have lived off that ammonium flux? How much biomass may it have sustained?

L331ff., Methods: Include what standard was used to determine instrument broadening.

Figure 1: Consider adding Fig. S1 as a small inset because it provides a quick idea on the origin of the cores geographically.

Figure 2: Specify what form of nitrogen in the legend and x axis label (bulk, organic, inorganic).

Figure 4: Adjust to “early Earth ocean” or “Hadean-Archean ocean”. Color the ocean in a brighter blue for better contrast (the color contrast is in general not optimal in this figure). “Along the path of seawater circulation into depths,..” – reword. Add more detail to the Box on the abiotic N₂ reduction.

Figure S8: Consider moving it into the main text, perhaps as additional panel in Fig. 2 or 3.

Reviewer #2

(Remarks to the Author)

The manuscript describes N-bearing secondary minerals in a hydrothermally altered basalt core of Oligocene age that are unusually isotopically depleted. These observations are novel and noteworthy, because isotope variation in the N system usually falls within a tighter range. The authors argue that the isotopically depleted nitrogen signatures were produced by the abiotic reduction of seawater-sourced N₂ by Fe(II) from the basaltic oceanic crust. They then make an effort to calculate the rate of reduction and its implications for transforming N₂ into bioavailable nitrogen by releasing NH₄⁺ into the Archean ocean.

In hydrothermal systems literature, studies examining nitrogen cycling are under-represented with respect to other biogeochemical systems (e.g., carbon and sulfur cycling are far better understood). These novel isotope data are supported by well-described abundance and mineralogical data. The methodological approach is strong. I do not doubt that these data will be of interest within the community.

However, the authors do not thoroughly evaluate other potential sources of nitrogen to the altered basalt core. In particular, they do not acknowledge the important role of NO₃⁻ reduction to NH₄⁺ in unsedimented seafloor hydrothermal systems. In nearly every modern study of NH₄⁺ isotopes in modern vent systems today, the abundance of NH₄⁺ expelled and its isotope signature can be fully attributed to the reduction of NO₃⁻ sourced from seawater. The authors miss an opportunity to acknowledge the existing literature on this topic, and they do not fully discount this pathway. We include more specific suggestions in the annotated paper attached to this review.

The authors emphasize that NH₄⁺ in modern vents is 'contaminated' by seawater-derived NH₄⁺, obscuring abiotic nitrogen reduction signals. This is not true for two reasons. First, seawater NH₄⁺ is quite low in abundance (~0.1 μmol/kg) compared with the range of NH₄⁺ abundance in unsedimented vent fluids (~1-40 μmol/kg). Seawater NH₄⁺ is very low compared to sediment-influenced vent fluids (10s of mmol/kg). Further, even if seawater-sourced NH₄⁺ was added to a fluid, it could be easily accounted for by isotope mass balance. It is well accepted that the majority of vent-sourced NH₄⁺ can be attributed to NO₃⁻ reduction.

Another important detail that is missing is whether or not different marine nitrogen cycling was different in the Oligocene ocean, which could contribute nitrogen sources that are isotopically distinct from modern N₂ and NO₃⁻, and therefore impact discussion of fractionation factors.

Finally, we were not convinced by the significance of the study to the Archean NH₄⁺ pool as presently described. The authors describe evidence for the abiotic production of NH₄⁺ that is co-precipitated with secondary hydrothermal minerals. The problem with this final calculation is that there is no discussion of how the NH₄⁺ would be released from secondary minerals into the Archean ocean. If it is immobilized by high temperature hydrothermal mineralization deep in the oceanic subsurface, then NH₄⁺ will not be released to seawater, nor will it be available biologically.

Reviewer #3

(Remarks to the Author)

Version 1:

Reviewer comments:

Reviewer #1

(Remarks to the Author)

My suggestions have been worked in and the manuscript presents itself significantly improved over last year's version. Just some minor comments below. I can recommend this updated manuscript for publication in Nature Communications.

L20. "at IODP Site U1502." Nature Communications has a very broad readership and so I suggest to change this to the general field site or geographical name (e.g., oceanic crusts in the South China Sea basin). The detail here is not needed in the abstract and virtually nobody will know what/where this is.

L56/57. "diminished surface NH₄⁺ in depths" – seems contradictory.. surface or deep NH₄⁺ ?

L177-180. These chemical formulas are much appreciated!

L230. just checking the grammar: "...may have been more active and productive.."?

Reference 4. It is Science volume 177 and not volume 117.

Reference 15. Check first author last name.

Reference 21. Missing publication year.

Figure 3. fonts are a bit small but I think the journal will still format this anyways.

Reviewer #4

(Remarks to the Author)

Review of "Abiotic nitrogen reduction in submarine hydrothermal systems: Implication to ammonium in prebiotic oceans" by WD Sun et al.

I am satisfied with this manuscript, with only several minor suggestions.

The authors conducted numerous analyses on altered basalts and hydrothermal veins from core U1502B, including investigations of mineral phases, analyses of major and trace elements, as well as measurements of N concentrations and isotopic ratios. The authors discovered that the hydrothermal veins have unusually high N concentrations and strongly ^{15}N -depleted isotopic signatures. The authors interpret the signatures as evidence of abiotic nitrogen reduction (ANC) from mantle-derived N_2 and use these data to estimate the global N flux generated by ANC.

Overall, this manuscript, including both experimental analyses and theoretical calculations, is robust, and the interpretation is reasonable. Also, the writing is reasonably good. Therefore, I recommend that this paper be accepted only with minor revisions.

The following suggestions aim to enhance the engagement of non-specialists and minimize potential misunderstandings.

1. The term "secondary silicate minerals" in Figure 2 may mislead some readers. Literally, quartz is a typical alteration mineral in this study, but it is not included in the "Altered silicate mineral" listed in Supplementary Table 1. I suggest this term should be revised for clarity.
2. As mentioned in the abstract of the manuscript, NH_3 or NH_4^+ can promote prebiotic reactions and resolve the faint young sun paradox. But many readers may not be familiar with the background involved. I suggest adding a relevant discussion.
3. I note that the other three reviewers provided numerous constructive comments, which have significantly contributed to improving the quality of this manuscript. I also find that the authors have provided thorough and thoughtful responses to these reviewers' suggestions. The authors have made much effort to improve the writing, incorporated additional thermodynamic calculations, and supplemented experimental data. I suggest that the sections on the added thermodynamic calculations and experimental supplements should be included in the supplementary information, which may help address potential questions from readers who have similar concerns.

REVIEWER COMMENTS

Reviewer #1 (Remarks to the Author):

Review of: Abiotic nitrogen reduction in submarine hydrothermal system

By: Sun et al.

Recommendation: Reconsider after revision.

Manuscript summary: The authors report on nitrogen isotopic variation in NH_4^+ in samples from an Oligocene mid-ocean ridge hydrothermal system and conclude that the extremely ^{15}N -depleted NH_4^+ observed in deep fluids, as recorded by vein minerals, can be best explained by abiotic reduction of mantle N_2 in the deep oceanic basement through the interaction between the deep fluids and Fe(II)-bearing minerals (e.g., olivine, pyroxene). Upscaling this process, they argue that this ammonium flux is small relative to the large nitrogen inventory in modern oceans, but could be an important contributor to quickly fertilize the early oceans and supply NH_3 to the atmosphere on the prebiotic Earth.

Review summary: Prior to the evolution of nitrogenases, there was essentially no direct pathway known to derive ammonium from N_2 (assuming the early Earth atmosphere was ammonia-poor). The abiotic reduction of N_2 through hydrothermal basalt has been hypothesized but direct evidence has been lacking. The study by Sun et al. delivers this evidence and therefore adds an important piece to our picture of the early nitrogen cycle.

Reply:

We appreciate the endorsement of the novelty and high quality of our study. The revisions corresponding to R1's comments (see details below) are highlighted in **brown** in the tracked version of the manuscript.

I find the logical flow of the investigation sound, the selected methods are robust, and the extrapolation of the findings are adequate. There are two major shortcomings that should be remedied before this manuscript could be reconsidered:

1) The writing has to be improved. In my specific comments I started out with language and grammar corrections but they became too extensive. Most notably, the last paragraph of the introduction reads like a Methods section. That last paragraph should define the question to be answered as well as tell the objective of the study which is not clear at all after reading the introduction. A more general reader might get highly confused. Moreover, the figures, especially the schematic, should be polished up.

Reply:

We have added a few sentences at the beginning and the end of the last paragraph in the Introduction to clarify the objectives of this study. Please see L65-67, L77-80.

We have also further improved the diagrams as suggested and added more detailed description in the captions for easy understanding.

2) The mechanism and catalytic reaction of N₂ reduction to ammonium by Fe(II)-bearing minerals has to be described more precisely. The authors should think more about the molecular mechanism that drives the reaction and suggest, if not identify, the Fe mineral serving as catalyst. Write out a proposed chemical reaction in a formula. The samples used are derived from an Oligocene bedrock, yet the authors draw conclusions to processes on the prebiotic Earth, well over 3 billion years ago. If the Fe minerals involved are better constrained, it can be argued that these minerals were also present in similar rock formations on the prebiotic Earth – which would essentially strengthen the extrapolation in time.

Reply:

1. Given that the hydrothermal system below the drilling site was sustained by intrusive dykes (Chen et al., 2023^{Ref 38}), the most likely Fe(II)-bearing mineral for the abiotic N₂ reduction is pyroxene (the rock constituting mineral of sheeted dykes and gabbros in oceanic crust) and possibly olivine (which can occur in small amounts in mafic dykes as well). This discussion and the chemical reactions have been added in L172-180.
2. These minerals are common in the early-Earth oceanic crust which was dominated by more (ultra)mafic lithology and experienced more serpentinization (Leong et al., 2021^{Ref 51} and reference therein), which could have induced more abiotic nitrogen reduction. This discussion has been added in L226-231.

Given the quality of the data and the logic that the authors applied to draw conclusions, I would recommend reconsidering this work for publication in Nature Communications after these two main points and the following minor points were reworked.

Specific comments:

Title: The wording of the title is awkward. The title should be better fleshed out and point towards the significance of the paper. For example: “Evidence for hydrothermal N₂ reduction to ammonium on the prebiotic Earth”.

Reply:

We appreciate for this appealing title. We have changed the title toward this direction as “*Abiotic nitrogen reduction in submarine hydrothermal systems: implication to ammonium in prebiotic oceans*”. See L2.

We did not directly adopt the suggested title because we try to avoid leading the readers to think we are studying hydrothermal samples from the prebiotic Earth in this study.

L14, Abstract: “... an essential component of the abiotic synthesis...”

Reply: revised as suggested. See L15.

L15, Abstract: take out “arguably” – such fill words are unnecessary.

Reply: revised as suggested.

L20, Abstract: name the nitrogen isotopes.

Reply: added as suggested. See L20, 21, 22.

Abstract, overall: I suggest including this key finding in the abstract. “Therefore, the extremely

^{15}N -depleted NH_4^+ in the deep fluids, as recorded by the vein minerals, can be best explained by abiotic reduction of mantle N_2 in the deep oceanic basement through the interaction between the deep fluids and Fe(II)-bearing minerals (e.g., olivine, pyroxene).”

Reply:

we would love to include this in the abstract. Unfortunately, due to the 150-word limit to the abstract of Nature Communications articles, we have to shorten the abstract significantly and have no room for this sentence.

L30-36. Split this very long sentence into two. Suggestion: “...laboratory experiments. These demonstrated that..”

Reply: revised as suggested. See L32.

L33: “early Earth surface”

Reply: revised as suggested. See L31.

L34ff.: References 7-9 on abiotic nitrogen reduction reactions on the early Earth are important but outdated. Please include more recent work:

Nishizawa, M. et al. Minerals 11, 321 (2021)

Buessecker, S. et al. Nat. Geosci. <https://doi.org/10.1038/s41561-022-01089-9> (2022)

Reply: added as suggested. Ref 14, 15 in the manuscript.

L40: “detecting an ANR signal...”

Reply: revised as suggested. See L52.

L51-57: Consider moving this to the methods and definitely modify the end of the introduction (see major comments).

Reply:

This paragraph is intended to include a brief introduction of the study site and hydrothermal vein samples (more detailed introduction was given in Methods and Supplementary Material), which are important justification for the representation and advantage of this set of samples to address the outlined outstanding questions. Therefore, we prefer to keep this information here.

Following the main comment above, we did add clear statement of goals of studying this set of samples (L65-67, 77-80), which should make the readers easier to follow.

L60-63: Clarify what N pool you are talking about – bulk, inorganic, organic?

Reply: It is bulk-rock N. This has been clarified in L84.

L99: use plural, “basements”

Reply: Revised as suggested. See L121.

L102, 103, and Fig. 1: Are these data (one data point) based on technical replicates? How was the heterogeneity in samples of the same core depth evaluated? The robustness especially of the measurement reporting -20.3% is important because it essentially creates the decreasing trend with depth. The authors should make sure this is not an outlier and a data point set of 3 such as for sample 21R would create more confidence.

Reply:

Yes, we did make duplicated measurements to validate the extremely low $\delta^{15}\text{N}$ value: two measurements gave: (1) $\text{N} = 179.0 \text{ ppm}$, $\delta^{15}\text{N} = -20.3\%$; (2) $\text{N} = 180.9$, $\delta^{15}\text{N} = -20.5\%$. We also did this practice of data validation on some other samples which could provide enough material for duplicate measurements. The duplicated data are now added to Supplementary Table 2.

It is worth clarifying that the 3 data from section 21R are 3 different samples rather than 3 repeated analyses of one sample. We were able to obtain this because section 21R has more abundant thick veins that allowed for sampling enough material for N isotope analysis. The sample availability in other sections is less mainly due to the veins in those sections are very thin so it is not feasible for sampling enough material for N isotope analysis.

L117-143: I like the logical flow of this paragraph very much. I think the way the conclusion that ^{15}N depletion in ammonium originates from reduction of mantle N_2 is derived is quite reasonable and sound.

Reply: Thanks again for the endorsement.

L134: Looking through the text, it is not explained where N_2 in the upper mantle actually comes from. The question of the ultimate source of N_2 would be important to evaluate how sustainable this reduced N source to the early oceans could have been. Please go into more detail (here or in the introduction) how N_2 in the upper mantle is thought to be derived and recycled.

Reply:

Nitrogen is known as a trace element (< 2 ppm) in the depleted mantle (the source of the MORB). Although the exact nitrogen species in the upper mantle is not well constrained, it has been widely observed that the N in the upper mantle is persistently exsolved and emitted as N₂ in mid-ocean ridges, with a global output flux at the same order of magnitude with the N₂ output flux at global arc volcanoes (Hilton et al., 2002^{Ref 45}; Bekaert et al., 2021^{Ref 46}). This huge amount of N₂ can provide a sustainable source for ANR. This discussion has been added in L163-166.

L158: Provide very briefly more insight into the locations and formation history where else significant 15N depletion was reported from. This will demonstrate better that the proposed process could be a global phenomenon.

Reply: added as suggested. See L198-201.

L173: I may have missed it somewhere in the supplements, but what is the basis of this calculated value. I suggest adding a reference to the supplements.

Reply: added as suggested. See L215, 217.

L183: How did serpentinization rate influence the GOE – please corroborate for readers not familiar with:

Leong, J.A.M., Ely, T. & Shock, E.L. Decreasing extents of Archean serpentinization contributed to the rise of an oxidized atmosphere. *Nat Commun* 12, 7341 (2021). <https://doi.org/10.1038/s41467-021-27589-7>

Reply: added as suggested. See L226-231 and Ref 51.

L191: Could an early biosphere have lived off that ammonium flux? How much biomass may it have sustained?

Reply: This is an intriguing question. Thinking about microbial system living on ammonium in the reducing early oceans, the mostly likely metabolic pathway would be sulfate-reducing anaerobic ammonium oxidation. We can look into this question from two different approaches:

1. In the laboratory microbe culture experiments carried out in anaerobic environment with NH₄⁺ and SO₄²⁻ in media, Mohammed Madani et al. (2022) observed fast growth of biomass in their experiments with pH = 5–10 (optimal at 8) and T = 20–50 °C (optimal at 30 °C), which well bracket the conditions of early-Earth oceans. The growth rate can be up to 7.3×10⁹ CFU/mL over 24-72 hour experiments, which is equivalent to 3.65×10¹² cells in the 500 ml flasks in their experiments. Although these experiments were carried out with a [SO₄²⁻] (~ 750 μM) slightly higher than the proposed [SO₄²⁻] in Archean oceans (2.5 – 200 μM; Habicht et al., 2002; Crowe et al., 2014), these experimental results clearly demonstrated that the sulfate and ammonium in the early oceans can support a biosphere, likely with less biomass yielded in the experiments.

2. From Gibbs energy point of view, we did a more quantitative assessment. We followed the calculation method of Schrum et al. (2009) for the sulfate-reducing ammonium oxidation process:

$$\Delta G_{(T,P,S)} = \Delta G^0_{(T,P,S)} + RT \ln \left(\frac{\gamma_{\text{N}_2}^4 [\text{N}_2]^4 \times \gamma_{\text{HS}^-}^3 [\text{HS}^-]^3 \times \gamma_{\text{H}^+}^5 [\text{H}^+]^5}{\gamma_{\text{NH}_4^+}^8 [\text{NH}_4^+]^8 \times \gamma_{\text{SO}_4^{2-}}^3 [\text{SO}_4^{2-}]^3} \right)$$

In which ΔG is a function of pressure (P), temperature (T) and salinity (S); ΔG^0 is the standard reaction Gibbs energy; γ is activity coefficient (see details in Schrum et al., 2009 and reference therein).

For a back-of-an-envelope calculation, we set the condition at fixed P of 1 bar and S of seawater salinity, but variable T of 15–70 °C, pH of 6–8, $[\text{SO}_4^{2-}]$ from 2.5–200 μM (Habicht et al., 2002; Crowe et al., 2014). The results are shown in Fig. R1 below. It shows that, under pH = 6–8, sulfate-reducing ammonium oxidation can release energy when $[\text{NH}_4^+]$ accumulates to ~200–4000 μM when $[\text{SO}_4^{2-}]$ is 200 μM or ~1000–15000 μM when $[\text{SO}_4^{2-}]$ is 2.5 μM . Such $[\text{NH}_4^+]$ is possible in our calculations since there was little sink for NH_4^+ in the prebiotic oceans. The energy yield enables to support a biosphere. However, due to the large uncertainty on the energy yield which is highly dependant on $[\text{NH}_4^+]$ and pH, it is difficult to constrain the biomass at the moment.

Fig. R1. Calculation of Gibbs energy for sulfate-reducing ammonium oxidation process

The above discussions have not been added into the manuscript yet. If the Editor and Reviewers think it is appropriate, we can add these results in the revision next round.

Reference:

- (1) Mohammed Madani, R. et al. Novel simultaneous removal of ammonium and sulfate by isolated *Bacillus cereus* strain from sewage treatment plant. *Water Air Soil Pollut.* **233**, 185 (2022).
- (2) Habicht, K.S., Gade, M., Thamdrup, B., Berg, P. & Canfield, D. E. Calibration of sulfate levels in the Archean ocean. *Science* **298**, 2372-2374 (2002).
- (3) Crowe, S. A. et al. Sulfate was a trace constituent of Archean seawater. *Science* **346**, 735-739 (2014).
- (4) Schrum, H. N., Spivack, A. J., Kastner, M. & D'Hondt, S. Sulfate-reducing ammonium oxidation: A thermodynamically feasible metabolic pathway in seafloor sediment. *Geology* **37**, 939-942 (2009).

L331ff., Methods: Include what standard was used to determine instrument broadening.

Reply: added as suggested. See L396-397.

Figure 1: Consider adding Fig. S1 as a small inset because it provides a quick idea on the origin of the cores geographically.

Reply: added as suggested. See Fig. 1.

Figure 2: Specify what form of nitrogen in the legend and x axis label (bulk, organic, inorganic).

Reply: revised as suggested. See Fig. 2.

Figure 4: Adjust to “early Earth ocean” or “Hadean-Archean ocean”. Color the ocean in a brighter blue for better contrast (the color contrast is in general not optimal in this figure). “Along the path of seawater circulation into depths,..” – reword. Add more detail to the Box on the abiotic N₂ reduction.

Reply:

1. We have revised the color and added details as suggested.
2. Figure caption has been re-worded.
3. This model is applicable to not only the Hadean-Archean oceans but Phanerozoic oceans as well, as it is established on the Cenozoic samples. So we still leave it as unspecified oceans.

Figure S8: Consider moving it into the main text, perhaps as additional panel in Fig. 2 or 3.

Reply: This is now added as Fig. 3c as suggested.

Reviewer #2 (Remarks to the Author):

The manuscript describes N-bearing secondary minerals in a hydrothermally altered basalt core of Oligocene age that are unusually isotopically depleted. These observations are novel and noteworthy, because isotope variation in the N system usually falls within a tighter range. The authors argue that the isotopically depleted nitrogen signatures were produced by the abiotic reduction of seawater-sourced N_2 by Fe(II) from the basaltic oceanic crust. They then make an effort to calculate the rate of reduction and its implications for transforming N_2 into bioavailable nitrogen by releasing NH_4^+ into the Archean ocean.

In hydrothermal systems literature, studies examining nitrogen cycling are under-represented with respect to other biogeochemical systems (e.g., carbon and sulfur cycling are far better understood). These novel isotope data are supported by well-described abundance and mineralogical data. The methodological approach is strong. I do not doubt that these data will be of interest within the community.

Reply: we appreciate the endorsement of the novelty of our study. The revisions corresponding to R2's comments (see details below) are highlighted in blue in the tracked version of the manuscript.

However, the authors do not thoroughly evaluate other potential sources of nitrogen to the altered basalt core. In particular, they do not acknowledge the important role of NO_3^- reduction to NH_4^+ in unsedimented seafloor hydrothermal systems. In nearly every modern study of NH_4^+ isotopes in modern vent systems today, the abundance of NH_4^+ expelled and its isotope signature can be fully attributed to the reduction of NO_3^- sourced from seawater. The authors miss an opportunity to acknowledge the existing literature on this topic, and they do not fully discount this pathway. We include more specific suggestions in the annotated paper attached to this review.

Reply:

we thank R2 to bring the PhD thesis of Charoenpong (2019) (now Ref 18 in our manuscript) to our attention. In this revision, we have added more discussions on (1) the diverse sources of hydrothermal fluid in the study site (Chen et al., 2023^{Ref 38}) and (2) the potential contribution of NO_3^- reduction to NH_4^+ in shallow hydrothermal systems, which actually strengthens our conclusion that the NH_4^+ from abiotic N_2 reduction in the deep hydrothermal fluid can be overprinted by NH_4^+ either derived from decay of organic matter, desorption from clays in sediments, and/or reduction of NO_3^- in seawater by biological or hydrothermal processes. Please see more explanations in the point-by-point replies below.

We also adopted the revision suggestions in the annotated PDF attachment and also itemized our reply below.

The authors emphasize that NH_4^+ in modern vents is 'contaminated' by seawater-derived NH_4^+ , obscuring abiotic nitrogen reduction signals. This is not true for two reasons. First, seawater NH_4^+ is quite low in abundance (~ 0.1 $\mu\text{mol/kg}$) compared with the range of NH_4^+ abundance in unsedimented vent fluids (~ 1 -40 $\mu\text{mol/kg}$). Seawater NH_4^+ is very low compared to sediment-influenced vent fluids (10s of mmol/kg). Further, even if seawater-sourced NH_4^+ was added to a fluid, it could be easily accounted for by isotope mass balance. It is well accepted that the majority of vent-sourced NH_4^+ can be attributed to NO_3^- reduction.

Reply*:

We would like to clarify that, the seawater-derived NH_4^+ in the last version of our manuscript was meant to refer to not just the low background NH_4^+ in seawater but the NH_4^+ derived from several nitrogen-bearing compounds in seawater, including those from NO_3^- reduction (Charoenpong, 2019^{Ref 18}), decay of organic matter, and desorption from clay minerals in sediments.

This has now been clarified by re-wording “seawater-derived NH_4^+ ” to “surface NH_4^+ ” with a detailed explanation of this term and the contributions from the multiple sources outlined above. See L36-51.

Another important detail that is missing is whether or not different marine nitrogen cycling was different in the Oligocene ocean, which could contribute nitrogen sources that are isotopically distinct from modern N_2 and NO_3^- , and therefore impact discussion of fractionation factors.

Reply**:

Recent studies indicate that the Oligocene-Miocene South China Sea had oxic bottom water which was well connected with the Pacific deep water (Zhang and Sun, 2023). Therefore, the Oligocene-Miocene nitrogen cycling in the South China Sea is expected not too different to those in the Pacific Ocean; the latter show a stable marine nitrogen cycle since the Oligocene as indicated by their nitrogen isotopic signatures that fall into the normal range of global marine sediments (+2‰ to +10‰; Sadofsky and Bebout, 2004; Kast et al., 2019).

We also obtained the bottom sediments from two IODP drill holes in the South China Sea, i.e., U1503A (~37 km south of our study site) and Site U1431E (~350 km southwest of our study site). They gave $\delta^{15}\text{N}$ values of +3.6‰ and +4.0‰, respectively, which also fall in the +2‰ to +10‰ range. Therefore, these sediment archives consistently show no significantly different marine nitrogen cycle in the Oligocene South China Sea.

So far, no evidence hints at significantly different marine nitrogen cycle since the Oligocene in the study area.

Reference:

- (1) Zhang, Z. & Sun, Z. The early-mid Miocene abyssal brown/green claystone from IODP Site U1503A in the northern South China Sea: Implications for paleoclimate and paleoceanography. *Gondwana Research* **120**, 286-303 (2023).
- (2) Sadofsky, S. J. & Bebout, G. E. Nitrogen geochemistry of subducting sediments: new results from the Izu-Bonin-Mariana margin and insights regarding global nitrogen subduction. *Geochem. Geophys. Geosyst.* **5**, Q03I15.
- (3) Kast, E.R., et al. Nitrogen isotope evidence for expanded ocean suboxia in the early Cenozoic. *Science* **364**, 386–389 (2019).

Finally, we were not convinced by the significance of the study to the Archean NH_4^+ pool as presently described. The authors describe evidence for the abiotic production of NH_4^+ that is co-precipitated with secondary hydrothermal minerals. The problem with this final calculation is that there is no discussion of how the NH_4^+ would be released from secondary minerals into the Archean ocean. If it is immobilized by high temperature hydrothermal mineralization deep in the oceanic subsurface, then NH_4^+ will not be released to seawater, nor will it be available biologically.

Reply***:

We would like to clarify that, NH_4^+ is a hydrophilic cation, geochemically close to Rb^+ . While the vein minerals can incorporate NH_4^+ from the fluid, it only accounts for a small fraction of the total NH_4^+ in fluid. The majority of the ANR-derived NH_4^+ would have remained in the fluid and been discharged to seawater when the deep fluid migrated to the surface. This can be more quantitatively demonstrated by the partition coefficient (D) of NH_4^+ between minerals and fluid, e.g., $D_{\text{Muscovite-fluid}} = 0.12$ and $D_{\text{K-feldspar-fluid}} = 0.14$ at 400 °C (Päer et al., 2004^{Ref 32} in our manuscript). These D values decrease quickly with decrease of temperature. Compared with muscovite and K-feldspar, the studied vein minerals here (plagioclase, epidote, chlorite) have even less NH_4^+ -hosting capabilities. For example, $D_{\text{Plagioclase-K-feldspar}} = 0.10-0.67$, $D_{\text{Plagioclase-Muscovite}} = 0.12-0.72$ (Honma & Itihara, 1981^{Ref 34}). Accordingly, the NH_4^+ content of fluid can be calculated by the NH_4^+ content of vein minerals and partition coefficient between minerals and fluid. The detailed calculation is given in Methods 2.4.

L17: check grammar here

Reply: We have reworded ‘sustainable amount of $\text{NH}_3/\text{NH}_4^+$ ’ to ‘sustainable $\text{NH}_3/\text{NH}_4^+$ ’. See L17.

L18: Grammar

Reply: Revised by removing “however”. See L18.

L33: Abiotic nitrogen reduction is a process, but I would not use the term reaction here because the authors have not specified the reactants.

Reply: Revised as suggested: ‘reaction’ is replaced by ‘process’. See L31.

L30-36: This sentence is very long. Suggest splitting it up to present each idea described.

Reply: Revised as suggested. The sentence is now split into two sentences. See L32.

L37-39: I do not agree that contamination by biogenic NH_4^+ in seawater is the reason for the lack of evidence for ANR in vent fluids.

Seawater NH_4^+ abundance is very low (~0.1 μM) compared to NH_4^+ abundance in vent fluids (~1-30 μM in unsedimented vents, ~10 μM in sedimented vents; see Supp Info in Reeves et al., PNAS, 2014 for a good list of NH_4^+ abundance data across many geologic settings). Contamination from seawater-derived NH_4^+ is a minor influence on the vent NH_4^+ pool.

Reply: Please see the Reply above marked by *, and the revision in L36-51.

A bigger factor is likely a relative lack of research into N cycling in modern hydrothermal systems in general, particularly in terms of the number of studies that combine abundance measurements with stable isotopes.

Reply: we totally agree with this point and added it in L46-49.

L42: Usually the assumption is that vent fluids are initially 100% deep seawater, rather than 'partially generated'

Reply:

We have removed “partially” to reduce confusion. Nevertheless, trace element and radiogenic isotope studies on the vein minerals from the studied drill core indicate that the hydrothermal fluid in the study site (U1502) involved not only modified seawater, but also magmatic fluid from the depths (Chen et al., 2023 Ref³⁸). This has been added in L75-77.

L45: This is an oversimplified statement. The earlier statement is true, that NH_4^+ in the circulating fluid can be incorporated into secondary minerals. However, the predominant thinking is that much of the NH_4^+ that vents at the seafloor is derived from the reduction of seawater-sourced NO_3^- .

The authors must acknowledge the possibility of seawater-sourced NO_3^- reduction as a source of NH_4^+ to circulating vent fluids, because NO_3^- is the second most abundant nitrogen species in seawater. It is particularly elevated in deep water.

Chapter 3 of the MIT/WHOI PhD thesis by Net Charoenpong has a great summary:

<https://darchive.mblwhoilibrary.org/bitstreams/20632fca-af49-5ee9-823b-0e4bc59e4a1c/download>

Reply:

As pointed out in the Reply above marked by *, we have acknowledged the contribution of NO_3^- in hydrothermal fluids. It should be pointed out that those literature data were mostly obtained from samples collected from hydrothermal vents, which belong to the ‘shallow hydrothermal fluid’ in our discussion and distinct to the ‘deep hydrothermal fluid’ we referred to in our manuscript.

L47: The mineral formulas do not contain NH_4^+ . It would be worth a brief comment here to state that NH_4^+ can substitute for K^+ (or other ions).

Reply: Revised as suggested. See L58-59.

L52: Given that NH_4^+ in hydrothermal systems is thought to be generated by reduction of seawater-derived NO_3^- , it would be worth stating whether the $\delta^{15}\text{N}$ of seawater NO_3^- or dissolved N_2 has remained constant since the Oligocene. Did the downwelling seawater in the Oligocene have the same N isotope characteristics as today?

This paper may be relevant:
https://www.science.org/doi/full/10.1126/science.aau5784?casa_token=WQhq182HUSEAAAAA%3Ai134L3BBliNljY89TB8VQKvc3PNjOxrB84ZgHkcYjffZU00OMr36gymvIKS2FO7Yd0kTWfKD_UsV-a5L

Reply: please see the Reply above marked by **.

L67: Have these basalts also experienced hydrothermal alteration?

Reply: Yes, numerous studies in elemental geochemistry and stable and radiogenic isotopes have demonstrated that these oceanic basalts have experienced hydrothermal alteration at various temperatures. The detailed sample description and relevant discussions with references have been given in the papers cited in this sentence. In our revision, we added 'hydrothermally altered basalts' to highlight this point. See L90-91.

L68: This ...observation?

Reply: We have rephrased this sentence for clarification. See L93-95.

L70: How high is high? How low is low?

Reply: The temperature ranges have been added. See L95-98.

L71: Can you split these ideas into shorter sentences and provide more specifics? Which type of mineral suite hosts more NH₄⁺? Briefly, why?

Reply: Revised as suggested. This information has also been incorporated in L93-98.

L83-86: This section is explained nicely and supported by the figures.

Reply: thanks.

L91-92: Check grammar here. Confusing phrasing.

Reply: the sentence has been rephrased; see L113-115.

L106-108: The logic works here - I can follow the thought process to explain the observation.

Reply: We appreciate this positive comment.

L109: By extremophiles, are you referring to living microbial biomass?

Reply: Yes. We replaced 'extremophiles' by 'living microbial biomass' to make it clear. See L131.

L111-114: These are very interesting observations, despite my critique of the analysis.

Reply: We appreciate the positive comment.

L119: Critically, the authors have not mentioned dissolved deepwater NO₃⁻. This must be included as a possible (and plausible) source of nitrogen, along with DON.

Reply: added as suggested. See L141-143.

L125: What if incomplete nitrate reduction to ammonium occurred at <200 °C in the shallow downwelling zone? It could then achieve more depleted (more negative values) and then be carried along with the fluid as it became progressing hotter, until being removed by secondary mineralization.

Chapter 2 of the MIT/WHOI PhD thesis by Net Charoenpong describes NO₃⁻ reduction experiments at hydrothermal conditions, including 150-200 degree experiments, that produce epsilon values between 11 to 67 per mil:

<https://darchive.mblwhoilibrary.org/bitstreams/20632fca-af49-5ee9-823b-0e4bc59e4a1c/download>

Reply:

1. We have carefully checked the experimental data in the PhD thesis by Charoenpong (2019), which has been referred in the discussion of N isotope fractionation factors in L150. We found that their results are in good agreement with our theoretical calculation results (Li Y. et al., 2021 ^{Ref 35}) except one point at a low temperature of 175 °C (see Fig. R2 below).
2. A detailed look at the data of the 175 °C experiments by Charoenpong (2019) show incomplete nitrogen yields, i.e., produced NH₄⁺ plus remaining NO₃⁻ are less than 100% (see Fig. R3a below). We speculate that the loss of nitrogen in these experiments may indicate that the nitrate reduction process in these experiments may not be simple nitrate reduction to ammonium but involve other gaseous byproduct such as NO and N₂O, which have been observed in similar experiments by Buessecker et al. (2022) ^{Ref 15}.
3. To further test this, we carried out abiotic nitrate reduction experiments in Fe²⁺-bearing solution at room temperature. The results also show significant N loss based on the mass balance of recovered NH₄⁺ product and remaining NO₃⁻ (see Fig. R3c). Therefore, the anomalously large isotope fractionations observed in both Charoenpong's 175 °C experiments and our room-T experiments (Fig. R3b, d) are superimposed effect of the isotope fractionation between NO₃⁻ and NH₄⁺ and the isotope fractionation between NO₃⁻ and NO or N₂O. A full understanding of this complexity requires much more experimental efforts which are beyond the scope of this study.
4. Nevertheless, despite this complexity, the temperatures of these experiments with anomalously large N isotope fractionations, i.e., 175 °C in Charoenpong (2019) or room-T in our new experiments, are all much lower than the formation T range of the studied vein samples (>200 °C). Thus the

fractionation factor of -67% obtained from the experiments is not considered as a reference datum for the discussion in the study.

Fig. R2. Comparison of N isotope fractionation factors between experimental results of Charoenpong (2019) and theoretical calculations of Li Y. et al. (2021).

Fig. R3. Comparison of N yield (a, c) and isotope fractionations (b, d) of abiotic nitrate reduction in laboratory experiments at $175\text{ }^{\circ}\text{C}$ (Charoenpong, 2019) and at room T ($21\text{ }^{\circ}\text{C}$; our unpublished data).

L126: do you mean less negative (more enriched?)

Reply: "higher" has been replaced by "less negative". See L150.

L141: Is Fe(II) the only reductant? What about aqueous H₂?

Reply:

Yes, H₂ could also be a potential reductant contributing to the process. But in the deep hydrothermal system in oceanic crust, H₂ is mainly produced by interaction of water and Fe²⁺-bearing minerals. Thus it is already included in the expression of "the interaction between the deep fluids and Fe²⁺-bearing minerals".

L143: See my earlier comment about the abundance of NH₄⁺ being very low in seawater. A more plausible NH₄⁺ source in the shallowest parts of the core could be biogenic marine sediment.

Reply:

To be consistent with the previous expression in the manuscript, it is now re-worded to 'surface NH₄⁺', which includes NH₄⁺ from a variety of surface processes, such as biogenic degradation, nitrate reduction, desorption from clay. See L171.

L151-152: Check grammar here.

Reply: the sentence has been rephrased. See L189-191.

L175: The flux is not into seawater, as the authors write, but rather into secondary minerals.

Reply: Please see the Reply above marked by ***.

L190-191: The NH₄⁺ will not reach the Archean ocean if it is fixed into secondary minerals deep in the seafloor by the processes the authors put forward. It might then reach the ocean if the minerals are subsequently weathered, but what if they are subducted instead? I don't buy this argument - how would the NH₄⁺ be mobilized, exactly?

Reply: Please also see the Reply above marked by ***.

L388: typo

Reply: corrected. See L448.

L495: Fluid compositions with Mg²⁺ = 0 are the product of a mathematical regression that completely removes seawater contamination. Not "little contamination"

Reply: Revised as suggested. See L554.

L501: What was the concentration of NH_4^+ and Rb in these two veins, and what average Rb fluid concentration was used, so that the reader can more easily follow the outcome of the calculation?

Reply: These values have been added. See L557-559.

Reviewer #3 (Remarks to the Author):

Response to Referees' comments

Reviewer #1 (Remarks to the Author):

My suggestions have been worked in and the manuscript presents itself significantly improved over last year's version.

Just some minor comments below. I can recommend this updated manuscript for publication in Nature Communications.

L20. "at IODP Site U1502." Nature Communications has a very broad readership and so I suggest to change this to the general field site or geographical name (e.g., oceanic crusts in the South China Sea basin). The detail here is not needed in the abstract and virtually nobody will know what/where this is.

Reply: revised as suggested (see L22-23).

L56/57. "diminished surface NH₄⁺ in depths" – seems contradictory.. surface or deep NH₄⁺ ?

Reply: the sentence has been rephrased for clarity (L59-60)

L177-180. These chemical formulas are much appreciated!

Reply: Thanks; no revision is needed here.

L230. just checking the grammar: "..may have been more active and productive.."?

Reply: "active and" is removed (see L234).

Reference 4. It is Science volume 177 and not volume 117.

Reply: Corrected (see L455).

Reference 15. Check first author last name.

Reply: Corrected (see L479).

Reference 21. Missing publication year.

Reply: I guess R1 meant Ref 19 & 20. The publication years have been added now. See L491, 493.

Figure 3. fonts are a bit small but I think the journal will still format this anyways.

Reply: Text in the diagram have been enlarged.

Reviewer #4 (Remarks to the Author):

The following suggestions aim to enhance the engagement of non-specialists and minimize potential misunderstandings.

1. The term “secondary silicate minerals” in Figure 2 may mislead some readers. Literally, quartz is a typical alteration mineral in this study, but it is not included in the “Altered silicate mineral” listed in Supplementary Table 1. I suggest this term should be revised for clarity.

Reply: For clarification, the term “secondary silicate minerals” should be “nitrogen-bearing secondary silicate minerals”. This has been corrected in both Figure 2 caption (L652) and Supplementary Table 1.

2. As mentioned in the abstract of the manuscript, NH_3 or NH_4^+ can promote prebiotic reactions and resolve the faint young sun paradox. But many readers may not be familiar with the background involved. I suggest adding a relevant discussion.

Reply: Added as suggested (see L33).

3. I note that the other three reviewers provided numerous constructive comments, which have significantly contributed to improving the quality of this manuscript. I also find that the authors have provided thorough and thoughtful responses to these reviewers’ suggestions. The authors have made much effort to improve the writing, incorporated additional thermodynamic calculations, and supplemented experimental data. I suggest that the sections on the added thermodynamic calculations and experimental supplements should be included in the supplementary information, which may help address potential questions from readers who have similar concerns.

Reply: As suggested, we have added the discussions based on thermodynamic calculations and incubation experiments into the Supplementary Information (text S3).